# Radiofrequency Irradiation Attenuated UVB-Induced Skin Pigmentation by Modulating ATP Release and CD39 Expression

**DOI:** 10.3390/ijms24065506

**Published:** 2023-03-14

**Authors:** Kyung-A Byun, Hyoung Moon Kim, Seyeon Oh, Kuk Hui Son, Kyunghee Byun

**Affiliations:** 1Department of Anatomy & Cell Biology, College of Medicine, Gachon University, Incheon 21936, Republic of Korea; 2Maylin Clinic, Ilsan 10391, Republic of Korea; 3Functional Cellular Networks Laboratory, Lee Gil Ya Cancer and Diabetes Institute, Gachon University of Medicine, Incheon 21999, Republic of Korea; 4Department of Thoracic and Cardiovascular Surgery, Gachon University Gil Medical Center, Gachon University, Incheon 21565, Republic of Korea

**Keywords:** radiofrequency, ultraviolet B, skin pigmentation

## Abstract

Hyperpigmentation stimulated by ultraviolet (UV)-induced melanin overproduction causes various cosmetic problems. UV radiation’s activation of the cyclic adenosine monophosphate (cAMP)-mediated cAMP-dependent protein kinase (PKA)/cAMP response element-binding protein (CREB)/microphthalmia-associated transcription factor (MITF) pathway is the main pathway for melanogenesis. However, the secretion of adenosine triphosphate (ATP) from keratinocytes due to UV radiation also leads to melanogenesis. Adenosine, converted from ATP by CD39 and CD73, can activate adenylate cyclase (AC) activity and increase intracellular cAMP expression. cAMP-mediated PKA activation results in dynamic mitochondrial changes that affect melanogenesis via ERK. We evaluated whether radiofrequency (RF) irradiation could decrease ATP release from keratinocytes and suppress the expression of CD39, CD73, and A_2A_/A_2B_ adenosine receptors (ARs) and the activity of AC and downregulate the PKA/CREB/MITF pathway, which would eventually decrease melanogenesis in vitro in UV-irradiated cells and animal skin. Our results indicate that RF decreased ATP release from UVB-irradiated keratinocytes. When conditioned media (CM) from UVB-irradiated keratinocytes (CM-UVB) were administered to melanocytes, the expressions of CD39, CD73, A_2A_/A_2B_ARs, cAMP, and PKA increased. However, the expression of these factors decreased when CM from UVB and RF-irradiated keratinocytes (CM-UVB/RF) was administered to melanocytes. The phosphorylation of DRP1 at Ser637, which inhibits mitochondrial fission, increased in UVB-irradiated animal skin and was decreased by RF irradiation. The expression of ERK1/2, which can degrade MITF, was increased using RF treatment in UVB-irradiated animal skin. Tyrosinase activity and melanin levels in melanocytes increased following CM-UVB administration, and these increases were reversed after CD39 silencing. Tyrosinase activity and melanin levels in melanocytes were decreased by CM-UVB/RF irradiation. In conclusion, RF irradiation decreased ATP release from keratinocytes and the expressions of CD39, CD73, and A_2A_/A_2B_ARs, which decreased AC activity in melanocytes. RF irradiation downregulated the cAMP-mediated PKA/CREB/MITF pathway and tyrosinase activity, and these inhibitory effects can be mediated via CD39 inhibition.

## 1. Introduction

The skin has protective roles as a barrier organ, as well as a neuro-endocrine function that stimulates the central nervous, endocrine, and immune systems to coordinate body homeostasis [1,2,3]. External stimuli, such as ultraviolet (UV) radiation, lead to the upregulation of various cytokines, urocortins, corticotropin-releasing hormone (CRH), proopiomelanocortin (POMC), and enkephalins, which further stimulates various skin responses, including melanogenesis [1,2,3,4].

Melanogenesis is a complex process that produces melanin in the melanocytes. The initial steps of melanogenesis involve the hydroxylation of phenylalanine to L-tyrosine [5,6,7]. L-tyrosine is then hydroxylated to L-dihydroxyphenylalanine (L-DOPA) by tyrosinase. Tyrosinase further oxidizes L-DOPA to L-DOPAquinone [5,6,7]. Depending on the presence of cysteine, L-DOPAquinone changes into either yellow-to-reddish pheomelanin or brown-to-blackish eumelanin [8,9,10]. Without cysteine, L-DOPAquinone is changed into DOPAchrome by tyrosinase-related protein (TRP)-1 and TRP-2, which is then further synthesized to eumelanin [8,9,10].

UV upregulates tumor suppressor protein p53, which further stimulates POMC. POMC is cleaved into adrenocorticotropic hormone (ACTH), α-melanocyte-stimulating hormone (MSH), β-MSH, and γ-MSH [11,12,13,14,15,16].

Secreted α-MSH binds to a melanocortin receptor (MC1R) in melanocytes, which promotes the dissociation of the α subunit and eventually upregulates the activity of adenylate cyclase (AC) [17]. ACs generate the second messenger cyclic adenosine monophosphate (cAMP) from adenosine triphosphate (ATP) [17]. cAMP leads to the increased activity of the cAMP-dependent protein kinase (PKA) and the transcription factor cAMP response element-binding protein (CREB), which sequentially increases the expression of microphthalmia-associated transcription factor (MITF) [18].

MITF is the main regulator of melanogenesis, modulating the survival, proliferation, and growth of melanocytes [19]. MITF also stimulates tyrosinase, TRP-1, and TRP-2 and eventually increases melanogenesis [19]. Aside from the cAMP-dependent pathway, Wnt/β-catenin, ERK/MAPK, and nitric oxide/cyclic guanosine monophosphate (cGMP) pathway lead to the upregulation of MITF in melanogenesis [20].

Extracellular adenosine regulates their G-protein-coupled receptors, which are classified into four types: A_1_, A_2A_, A_2B_, and A_3_ [21]. A_1_ and A_3_ adenosine receptors (ARs) decrease the activity of AC and lead to the inhibition of cAMP generation; however, A_2A_ and A_2B_ARs lead to the activation of AC, which increases intracellular cAMP [22,23].

cAMP-mediated PKA activation, which is known to activate CREB, is also involved in mitochondrial dynamics [24,25,26]. During the mitochondrial fission process, dynamin-related protein 1 (DRP1) moves to the surface of mitochondria and is involved in dividing mitochondria [27]. PKA leads to the phosphorylation of DRP1 at Ser656, which inhibits mitochondrial fission [28]. Moreover, PKA-induced DRP1 phosphorylation at Ser637 inhibits the translocation of DRP1 to the mitochondrial surface and thus results in mitochondrial elongation [29,30]. However, the dephosphorylation of DRP1 restored the recruitment of DRP1 to mitochondria, which caused increased fission [28,31,32].

UV radiation results in mitochondrial elongation and increased melanogenesis [33]. Moreover, the genetic deletion of DRP1 leads to increased expressions of TRP-1 and MITF [33]. In contrast, increased mitochondrial fragmentation via the inhibition of optic atrophy type 1 (OPA1), which is involved in mitochondrial fusion, leads to decreased melanogenesis [33]. In addition, ERK1/2 activation induces the proteasomal degradation of MITF and thus decreases melanogenesis [34,35], and mitochondrial fission leads to the activation of ERK1/2 [36]. Therefore, mitochondrial dynamics, such as decreased mitochondrial fission, are involved in melanogenesis via ERK1/2 [33].

Thus, ATP release from UV-irradiated keratinocytes could be involved in increasing intracellular cAMP levels in melanocytes via CD39/CD73 and A_2A_ or A_2B_ARs. Since the cAMP-mediated activation of CREB and MITF is the main pathway of melanogenesis [17], we believe that UV-induced ATP release from keratinocytes could affect melanogenesis via CD39/CD73.

Radiofrequency (RF) is a form of electromagnetic energy that generates molecular agitation in tissues [37]. Moreover, a conducted electric current caused by RF is transformed into heat [37]. Since the heat generated by RF stimulates collagen synthesis by increasing the expression of various heat shock proteins (HSPs), such as HSP47 and HSP70, RF has been used for skin rejuvenation [38,39].

Previously, our group reported that RF irradiation decreased skin pigmentation by suppressing melanogenesis via the modulation of HSP70 [40] and increased melanin removal by elevating autophagy [41].

Although it is known that UV radiation can induce ATP release from keratinocytes and lead to melanogenesis in melanocytes, it has not yet been revealed whether RF irradiation can decrease melanogenesis by modulating ATP release.

We hypothesized that RF irradiation decreases ATP release from keratinocytes and the expressions of CD39, CD73, A_2A_AR, and A_2B_AR, which then suppress the activity of ACs. Decreased AC activity leads to the downregulation of cAMP generation and PKA, which eventually decreases CREB expression and increases mitochondrial fission. Decreased CREB expression and increased mitochondrial fission lead to decreased MITF levels, which eventually decreases melanogenesis. We evaluated the effect of RF irradiation on melanogenesis by decreasing ATP release from keratinocytes using an in vitro model of UVB-irradiated keratinocytes and a UVB-irradiated animal model.

## 2. Results

### 2.1. RF Irradiation Decreased ATP Release from UV-Radiated Keratinocytes

First, we evaluated whether RF irradiation decreased ATP release from UVB-exposed keratinocytes. After human keratinocytes were subjected to UVB radiation, RF irradiation was performed. Forty-eight hours after RF irradiation, ATP levels in the supernatant of keratinocytes were measured (Appendix A). The ATP level was significantly increased by UVB radiation and significantly decreased by RF treatment (Figure 1A).

### 2.2. RF Decreased the Expression of CD39, CD73, and the A_2A_/A_2B_ARs in Melanocytes and UV-Irradiated Animal Skin

Next, we evaluated whether RF irradiation decreased the expressions of CD39, CD73, and A_2A_/A_2B_ARs in melanocytes by decreasing ATP release from keratinocytes. The conditioned media (CM) obtained from CON-, UVB-, or UVB and RF-irradiated keratinocytes were used to treat human melanocytes (Appendix A).

The expression of CD39 and CD73 in melanocytes was significantly increased with the treatment of CM from UVB-irradiated keratinocytes (CM-UVB) and significantly decreased with the treatment of CM from UVB- and RF-irradiated keratinocytes (CM-UVB/RF) (Figure 1B–D).

The expression of the A_2A_/A_2B_ARs in melanocytes significantly increased under treatment with CM-UVB and significantly decreased under treatment with CM-UVB/RF (Figure 1B,E,F).

The expression of CD39 and CD73 was also increased by UVB radiation in the skin of animals subjected to UVB radiation compared with that of control animals that did not receive UVB radiation. The expressions of CD39 and CD73 were significantly decreased by RF irradiation (Figure 2A–C).

Moreover, the expression of A_2A_/A_2B_ARs was increased in the skin of animals subjected to UVB radiation and decreased by RF irradiation (Figure 2A,D,E).

### 2.3. RF Irradiation Decreased AC Activity, cAMP Levels, and PKA Expression in Melanocytes and Animal Skin

Next, we evaluated whether RF irradiation decreased AC activity in melanocytes by modulating ATP release from keratinocytes.

The activity of AC was increased in melanocytes via the administration of CM-UVB and was significantly decreased by CM-UVB/RF treatment (Figure 3A).

The level of cAMP was increased in melanocytes via the administration of CM-UVB and was significantly decreased by CM-UVB/RF treatment (Figure 3B).

The expression of PKA was increased in melanocytes via the administration of CM-UVB, and PKA expression was significantly decreased by CM-UVB/RF treatment (Figure 3C).

The results suggest that RF irradiation can modulate melanocytes to decrease the activity of AC and cAMP and the expression of PKA via the culture supernatant of keratinocytes.

In animal skin, the activity of AC, level of cAMP, and expression of PKA were increased by UVB radiation and decreased by RF irradiation (Figure 3D–G).

### 2.4. RF Irradiation Decreased the Expression of CREB, Phosphorylation of DRP1, and Expression of MITF

The expression of CREB in UVB-irradiated mouse skin was significantly increased, and it was decreased by RF irradiation (Figure 4A).

UVB radiation increased the phosphorylation of DRP1 at Ser637 in animal skin, which was then decreased by RF irradiation (Figure 4B,C).

The expression of OPA1 and mitofusin 2 (MFN2), which are mitochondrial fusion markers, was increased by UVB radiation and decreased by RF irradiation (Figure 4B,D,E).

The expression ratio of pERK1/2/ERK1/2 was decreased by UVB radiation and increased by RF irradiation (Figure 4F,G).

Additionally, pMITF and MITF expression levels increased in the skin of animals subjected to UVB radiation and decreased by RF irradiation (Figure 4H–J).

### 2.5. RF Irradiation Decreased Tyrosinase Activity and UVB-Induced Melanin Accumulation

Next, we evaluated whether RF irradiation decreased melanin synthesis. Tyrosinase activity in melanocytes was increased by the administration of CM-UVB. Upon silencing CD39, tyrosinase activity was not increased by the administration of CM-UVB compared with the CON. The increased tyrosinase activity caused by the administration of CM-UVB was significantly decreased by CM-UVB/RF treatment. However, among silencing CD39 melanocyte, tyrosinase activity was not different when either CM-UVB or CM-UVB/RF were treated (Figure 5A).

The melanin content in melanocytes was increased by the administration of CM-UVB. Upon silencing CD39, melanin content was not increased by the administration of CM-UVB compared with that of the CON. The increased melanin content by administration of CM-UVB was significantly decreased by CM-UVB/RF administration. However, among silencing CD39 melanocyte, melanin content was not different following treatment by either CM-UVB or CM-UVB/RF (Figure 5B).

The tyrosinase activity in the UVB-irradiated animal skin increased, and it was decreased by RF irradiation (Figure 5C). Melanin accumulation in the dermis was evaluated with Fontana Masson staining. The staining results show that melanin accumulation in the dermis was increased by UVB radiation and decreased by RF irradiation (Figure 5D,E).

## 3. Discussion

Melanin in the skin, as a photoprotective pigment, absorbs UV and acts as scavenger against reactive oxygen/nitrogen species (ROS/RNS) [42]. Proper melanogenesis after UV exposure is essential for skin protection; however, the overproduction of melanin causes various cosmetic problems, such as senile lentigines, post-inflammatory hyperpigmentation, freckles, and dots [43].

Recently, ATP was reported to be involved in melanogenesis [44]. UVB radiation leads to increased ATP release from keratinocytes and human skin. The administration of ATP to melanocytes leads to the increased expression of tyrosinase, CREB, and MITF [44]. ATP is a ligand of P2 receptors that is involved in intracellular Ca^2+^ mobilization [45], and ATP-induced melanogenesis is mediated by the P2X7 receptor [44].

Extracellular ATP is changed into adenosine by serial reactions with CD39 and CD73 [46]. By binding with the A_2A_ or A_2B_AR, adenosine increases AC activity, which increases intracellular cAMP expression [22,23].

The purpose of our study was to evaluate whether RF could decrease UVB-induced melanogenesis via decreasing ATP release from keratinocytes. Thus, we designed an in vitro model in which CM from UVB-irradiated keratinocytes was administered to melanocytes. First, we sought to evaluate whether UV radiation increased ATP release from keratinocytes. ATP levels in the supernatant from UVB-irradiated keratinocytes increased compared with those of control keratinocytes (Figure 1A). Next, we evaluated how elevated ATP secretion from keratinocytes affects melanocytes to increase melanogenesis. Upon treating melanocytes with CM from UVB-irradiated keratinocytes, the expressions of CD39, CD73, and A_2A_/A_2B_ARs increased (Figure 1B–F). Moreover, AC activity and cAMP levels were increased by CM from UVB-irradiated keratinocytes. It seemed that UVB radiation increased ATP release from keratinocytes, and released extracellular ATP was converted into adenosine by CD39 and CD73, which stimulated A_2A_/A_2B_ARs. The upregulation of A_2A_/A_2B_ARs led to increased AC activity and increased cAMP levels in melanocytes (Figure 3A,B). Moreover, increased cAMP was accompanied by an increased expression of PKA in melanocytes treated with CM from UVB-irradiated keratinocytes (Figure 3C).

Our results also show increased tyrosinase activity and melanin contents in melanocytes via treatment with CM from UVB-irradiated keratinocytes. When CD39 was silenced by siCD39, tyrosinase activity and melanin content were not increased by treatment with CM from UVB-irradiated keratinocytes (Figure 5A,B). This result suggests that melanogenesis triggered in melanocytes by ATP secretion from keratinocytes could be mediated by CD39. To the best of our knowledge, this is the first study that shows the role of CD39 and ARs in UV radiation-induced melanogenesis. UV radiation also led to increased expressions of CD39, CD73, and A_2A_/A_2B_ARs, as well as AC activity, cAMP levels, and PKA expression in animal skin (Figure 2 and Figure 3D–G). Moreover, melanogenesis-related factors such as CREB and MITF were increased in UV-irradiated animal skin (Figure 4A,H,I).

RF irradiation decreases melanogenesis by increasing HSP70, which inhibits p53, MC1R, CREB, and MITF in the skin of animals subjected to UVB radiation [40]. We sought to evaluate whether RF irradiation could modulate ATP release from UVB-irradiated keratinocytes, which increases melanogenesis in melanocytes. RF irradiation decreased ATP levels in keratinocytes subjected to UVB radiation. Moreover, the expressions of CD39, CD73, A_2A_/A_2B_ARs, AC activity, cAMP levels, and PKA expression in melanocytes treated with CM from UVB- and RF-irradiated keratinocytes were lower than those in melanocytes treated with CM from UVB-irradiated keratinocytes. Tyrosinase activity in melanocytes was decreased by CM from UVB- and RF-irradiated keratinocytes compared with those of UVB-irradiated keratinocytes. However, among silencing the CD39 melanocyte, tyrosinase activity was not different when treated with either CM-UVB or CM-UVB/RF.

Those results show that RF irradiation decreased ATP release from keratinocytes, and these decreased ATP levels affected melanogenesis in melanocytes. In this study, we did not evaluate the mechanism by which RF irradiation decreased ATP release in keratinocytes. Thus, the mechanism of this finding should be evaluated in future studies.

Mitochondrial dynamics, which affect various cell functions, are involved in determining cell death or survival [47,48]. It was reported that UV radiation induced mitochondrial dysfunction, which led to cellular damage [49]. Moreover, mitochondrial dynamics are related to melanogenesis.

The loss of prohibition triggered by 3-isobutyl-1-methylxanthine (IBMX), which promotes mitochondrial homeostasis, resulted in an increased mitochondrial fission and thus decreased melanogenesis [50,51]. The UV irradiation of B16F1 mouse melanoma cells at 0.5 and 1 J/cm^2^ led to mitochondrial elongation and melanogenesis [33]. Since the phosphorylation of DRP1 at Ser637 via cAMP-mediated PKA activation is known to downregulate DRP1 and eventually lead to mitochondrial elongation [29], we evaluated whether UV-induced DRP1 phosphorylation at Ser637 modulates mitochondrial dynamics via cAMP-mediated PKA activation. UV-irradiated animal skin had an increased expression of phosphorylated DRP at Ser637, and RF irradiation decreased this expression (Figure 4B,C). We did not directly evaluate mitochondrial morphology using any imaging technique. However, mitochondrial fusion-related proteins such as OPA1 and MFN2 were increased by UV radiation in animal skin and decreased by RF irradiation (Figure 4B,D,E). Although we did not observe mitochondrial elongation with imaging, we could infer that increased phosphorylation of DRP at Ser637 affects mitochondrial dynamics.

Since mitochondrial fission activates ERK and degrades MITF [33], we also evaluated ERK changes after RF irradiation. UVB radiation decreased the pERK1/2/ERK1/2 ratio, which was increased by RF irradiation (Figure 4F,G).

Accompanied by ERK upregulation, the expression of MITF and tyrosinase activity in the skin of animals subjected to UVB radiation was decreased by RF treatment. Melanin accumulation in UV-irradiated animal skin was also decreased by RF irradiation.

The cosmetic industry has performed extensive research to treat hyperpigmentation-related problems using various methods, such as cosmetics, chemicals, drugs, or medical devices [52,53,54]. RF therapy could be another possible tool to solve cosmetic problems caused by UV-induced hyperpigmentation.

In conclusion, RF irradiation decreased ATP release from keratinocytes, the expressions of CD39, CD73, and A_2A_/A_2B_ AR and the activity of AC in melanocytes, which ultimately decreased cAMP levels. As a result, the expressions of PKA and CREB also decreased. The phosphorylation of DRP1 at Ser637, which affects mitochondrial dynamics, was decreased by RF irradiation. This irradiation also decreased tyrosinase activity and melanin synthesis, but these decreasing effects were not observed when CD39 was inhibited (Figure 5F).

## 4. Materials and Methods

### 4.1. Radiofrequency Irradiation System

A radiofrequency device (POTENZA, Jeisys, Seoul, Republic of Korea) was used in this study. It has an impedance checking and feedback system which was used to determine the compensation value by automatically measuring impedance. The most common configurations of electrode systems are monopolar, bipolar, and multipolar, including fractional systems where this effect is achieved by the superposition of RF current paths between paired electrodes. Bipolar radiofrequency energy was delivered to the skin surface and epidermis via a noninvasive electrode tip, SFA tip. The tip consisted of 192 electrodes with a diameter of 0.2 mm arranged at a spacing of 1 mm, and it is designed to deliver RF energy at a shallow depth. The RF conditions were 10 W, 100 ms, and 2 pulses. Additionally, the frequency was 2 MHz, and the wavelength is inversely proportional to the frequency [55].

### 4.2. In Vitro Model

#### 4.2.1. RF Irradiation in UVB-Exposed Keratinocytes

HEKn (ATCC, Manassas, VA, USA) were cultivated in a Dermal Cell Basal Medium (ATCC) containing a keratinocyte growth kit (ATCC) in an incubator at 37 °C with 5% CO_2_. When HEKn reached 70% confluence, RF (2 MHz, 10 W/100 ms) was applied after exposure to UVB (200 mJ/cm^2^) for 5 min. UVB used an instrument with a peak wavelength of 306 nm (G15T8E; SANKYO DENKI, Yokohama, Japan). RF-treated HEKn were cultured in an incubator at 37 °C with 5% CO_2_ for 48 h, and the cell lysate and supernatant (conditioned media, CM) were obtained (Appendix A).

#### 4.2.2. CM Treatment in Melanocytes

HEMn (ATCC) were grown in Dermal Cell Basal Medium (ATCC) with a melanocyte growth kit (ATCC) in a 37 °C/5% CO_2_ incubator. When HEMn confluence reached 80% in the dish, the mixture of CM obtained in Section 4.2.1. and standard media (growth media; GM) was used to treat HEMn for 48 h, and the cell lysate was collected (Appendix A).

#### 4.2.3. Silencing of CD39 in Melanocytes

When HEMn confluence reached 80% in the dish, transfection with CD39 shRNA (Santa Cruz Biotechnology, Dallas, TX, USA) was conducted by Lipofectamine 3000 reagent (Invitrogen, Carlsbad, CA, USA) according to the manufacturer’s protocol. After transfection, HEMn was treated as described in Section 4.2.2 (Appendix A).

#### 4.2.4. Measurement of Melanin Content in Cells

To assess the melanin content of HEMn, the cells were seeded at 1 × 10^4^ cells/well in 96-well plates and incubated for 24 h. After treatment with HEKn supernatant, the cells were harvested via centrifugation at 12,000× *g* for 20 min and lysed in 100 μL of 10% di-methyl sulfoxide and 1 NaOH. After incubation in NaOH solution at 95 °C for 20 min, the absorbance at 490 nm was measured.

### 4.3. Skin Pigmentation Model

HRM-2 mice hairless mice (6 weeks old, male, 20–25 g), capable of synthesizing melanin, were obtained from Japan SLC, Inc. (Shizuoka, Japan), and underwent acclimatization for 2 weeks. The mice were housed in cages under a 12 h light/dark cycle with a controlled temperature of approximately 23 °C, relative air humidity of approximately 50%, and ad libitum access to food and water.

After the adaptation period, the mice were randomly separated into three groups as follows:(1)CON (no exposure to UVB with no irradiated RF);(2)UVB (exposure to UVB at 200 mJ/cm^2^ with no irradiated RF);(3)UVB/RF (exposure to UVB at 200 mJ/cm^2^ with irradiated RF).

The mice were exposed to UVB for 5 min once every 2 d for 10 d and then for 5 min every day for 3 d (total of 13 d). Subsequently, the mice were irradiated with RF and exposed to UVB every 2 d for 28 d (Appendix A).

This study was approved by the Center of Animal Care and Use Ethical Board of Gachon University (Approval Number LCDI-2020-0115) and executed in accordance with the Institutional Animal Care and Use Committee.

### 4.4. Protein Sample Preparation

Protein extraction (EzRIPA lysis kit; ATTO, Tokyo, Japan) was performed according to the manufacturer’s instructions. Briefly, 100 mg of frozen tissue was cut into small pieces, 1 mL of RIPA buffer containing protease inhibitor and phosphatase inhibitor was added, and then the tissue was homogenized using a Bioprep-24R instrument (Allsheng, Hangzhou, China). Then, the samples were incubated on ice for 15 min and mixed once more with sonication. Finally, centrifugation was performed at 14,000× *g* at 4 °C for 15 min, and the supernatant was transferred to a new tube. Then, protein quantification was confirmed through bicinchoninic acid assay (Thermo Fisher Scientific, Waltham, MA, USA).

### 4.5. Assay

#### 4.5.1. ATP Release

An ATP release assay (ATP assay kit; Abcam, Cambridge, UK) was performed according to the manufacturer’s instructions. Briefly, standard and keratinocyte were aliquoted in equal amounts in 96-well plates. Reaction mix solution was prepared by mixing the reagents included in the kit, and 50 μL was dispensed into each well. The sample and reaction mix were mixed and incubated for 30 min at room temperature protected from light. The absorbance was measured at 570 nm.

#### 4.5.2. Adenylate Cyclase

An adenylate cyclase assay (ADCY kit; Mybiosource, Vancouver, BC, Canada) was performed according to the manufacturer’s instructions. Briefly, 50 μL of standard and sample was dispensed in a 96-well plate, and 50 µL of the conjugate reagent included in the kit was added and incubated at 37 °C for 1 h. After that, the cells were washed 5 times with wash buffer for 10 min each, 50 μL of substrate A and 50 μL of substrate B were added, and the mixture was incubated at 37 °C for 15 min in the dark. After that, 50 μL of stop solution was added, and the absorbance was measured at 450 nm.

#### 4.5.3. Cyclic AMP

A cyclic AMP assay (cAMP XP assay kit; Cell Signaling, MA, USA) was performed according to the manufacturer’s instructions. Briefly, after preparing cAMP standards, the samples and standards were aliquoted in equal amounts in 96-well plates. Fifty microliters of cAMP-HRP conjugate reagent included in the kit was dispensed and incubated for 3 h at room temperature. After washing 4 times with wash buffer, 100 μL of TMB substrate was added and incubated for 30 min at room temperature. Then, 100 μL of stop solution was added, and the absorbance was measured at 450 nm.

#### 4.5.4. Tyrosinase Activity

A tyrosinase activity assay (Tyrosinase activity assay kit; Abcam, Cambridge, UK) was performed according to the manufacturer’s instructions. Briefly, after dispensing the same amount of sample in a 96-well plate, 50 μL of the reaction mix included in the kit was added to the wells and thoroughly mixed, and the absorbance was measured at 510 nm in 30 s intervals during the reaction at 37 °C.

### 4.6. Western Blot

Total protein (30–50 µg) was loaded on an 8–12% polyacrylamide gel and separated using electrophoresis (Criterion System, Bio-Rad Laboratories, Inc., Hercules, CA, USA). The separated protein was transferred to a PVDF membrane and blocked with 5% skim milk. Thereafter, the cells were incubated overnight with a primary antibody listed in Appendix A, washed with tris-buffered saline with 0.1% tween 20, and incubated with a secondary antibody according to the host. The proteins were visualized with an enhanced chemiluminescence substrate (Cytiva, Vancouver, BC, Canada) on a digital acquisition system (Bio-Rad, CA, USA). Individual protein expression values were quantified using ImageJ software (National Institutes of Health, Maryland, MD, USA), and differences in protein expression were normalized to the values of β-actin and expressed as relative to the mean of the CON group.

### 4.7. Quantitative Real-Time Polymerase Chain Reaction

#### 4.7.1. Extraction of RNA and cDNA Synthesis

RNA extraction (RNAiso Plus; Takara, Shiga, Japan) was performed according to the manufacturer’s instructions. Fifty milligrams of frozen tissue were cut into small pieces, 1 mL of RNAiso Plus was added, and then the tissue was homogenized with a Bioprep-24R instrument (Allsheng, Hangzhou, China). Then, the cells were incubated at room temperature for 5 min and centrifuged at 12,000× *g* at 4 °C for 5 min, and the supernatant was transferred to a new tube. After, 0.2 mL of chloroform was added, vortexed, incubated at room temperature for 5 min, and centrifuged 12,000× *g* at 4 °C for 15 min. Then, the supernatant was transferred to a new tube and 0.5 mL of isopropanol was added. The solution was mixed, incubated at room temperature for 10 min, and centrifuged 12,000× *g* at 4 °C for 10 min. After, the RNA was washed with 1 mL of 75% ethanol, and centrifuged 7500× *g* at 4 °C for 5 min. Finally, the supernatant was discarded and the pellet was maintained and dissolved with diethyl pyrocarbonate-treated water.

cDNA synthesis (PrimeScript™ 1st strand cDNA Synthesis Kit; Takara, Shiga, Japan) was performed according to the manufacturer’s instructions. Briefly, the extracted RNA was quantified and converted to cDNA for quantitative real-time polymerase chain reaction (qRT-PCR).

#### 4.7.2. Quantitative Real-Time Polymerase Chain Reaction

SYBR Green reagent was mixed with 1 μg of the synthesized cDNA template and 10 pmol of the primers (Appendix A) was dispensed into a 384-well multiplate. The mixture was then analyzed using a CFX386 Touch Real-Time PCR System (Bio-Rad, Hercules, CA, USA).

### 4.8. Preparation of Paraffin-Embedded Tissue

Skin tissue was fixed with cold 4% paraformaldehyde (Sigma-Aldrich, St. Louis, MO, USA). The fixed tissue samples were washed for 30 min for embedding, and a tissue processor (Thermo Fisher Scientific) was used to generate a paraffin block of skin tissue. Paraffin blocks were cut to 7 μm using a microtome (Leica, Wetzlar, Germany) and dried at 60 °C for 24 h for mounting to coated slides (MUTO PURE CHEMICALS CO., LTD., Tokyo, Japan).

### 4.9. Fontana-Masson Stain

Fontana-Masson staining (Fontana-Masson Stain Kit; ScyTek, Logan, UT, USA) was performed according to the manufacturer’s instructions. Briefly, the slides were deparaffinized, and the skin tissue was incubated at 60 °C for 30 min in Fontana ammonia silver solution (ScyTek, West Logan, UT, USA). Then, after rinsing 3 times with distilled water, the dyed areas other than melanin were removed with 0.2% gold chloride solution and 5% sodium thiosulfate solution. Thereafter, nuclear staining was performed with Nuclear Fast Red Solution stain, followed by a dehydration process, and then a cover slide was mounted and observed under a microscope.

### 4.10. Statistical Analysis

We performed Kruskal–Wallis tests to compare the three groups, followed by a Mann–Whitney U test for post hoc comparisons. This study was validated using an unpaired t test. All the results are presented as the mean ± standard deviation, and statistical analyses were performed using SPSS v.22 (IBM Corporation; Armonk, NY, USA).

## Figures and Tables

**Figure 1 ijms-24-05506-f001:**
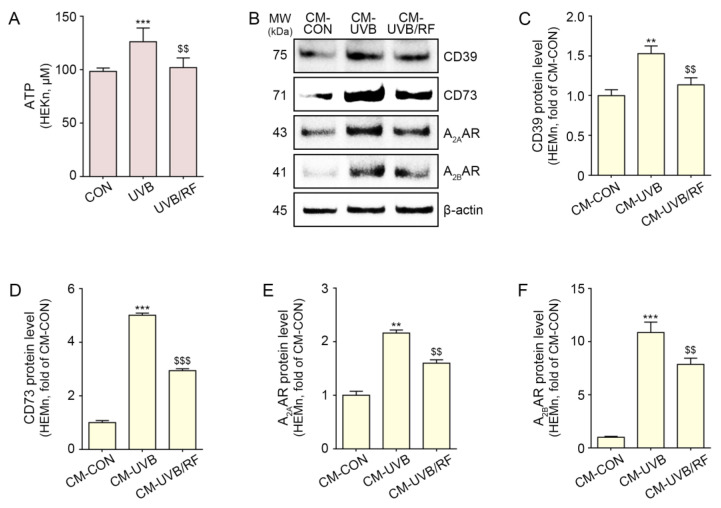
CD39/CD73/A_2A_AR/A_2B_AR reduction in melanocytes through regulation of ATP release from UVB/RF-exposed keratinocytes. HEKn were not exposed or exposed to UVB for 5 min (CON or UVB group), and then RF was irradiated after exposure to UVB (UVB/RF group). (**A**) ATP release in the CON group, UVB group, and UVB/RF group was verified using an ATP release assay. (**B**–**F**) HEMn were treated with a supernatant of HEKn (CM-CON), UVB-exposed HEKn (CM-UVB), or UVB and RF-treated HEKn (CM-UVB/RF) for 48 h. (**B**) The protein expression of CD39/CD73/A_2A_AR/A_2B_AR in CM-treated HEMn was measured by Western blot. (**C**–**F**) Quantitative graph for Western blot of (**B**). Data are presented as the mean ± SD of three independent experiments. **, *p* < 0.01; ***, *p* < 0.001, second bar vs. first bar; $$, *p* < 0.01, $$$, *p* < 0.001 third bar vs. second bar (Mann–Whitney U test). A_2A_AR, A_2A_ adenosine receptor; A_2B_AR, A_2B_ adenosine receptor; ATP, adenosine triphosphate; CD39, ectonucleoside triphosphate diphosphohydrolase-1; CD73, 5′-nucleotidase; CM, conditioned media; CON, control; HEKn, human epidermal primary keratinocytes; h, hours; HEMn, human epidermal primary melanocytes; MW, molecular weight; RF, radiofrequency; UVB, ultraviolet B.

**Figure 2 ijms-24-05506-f002:**
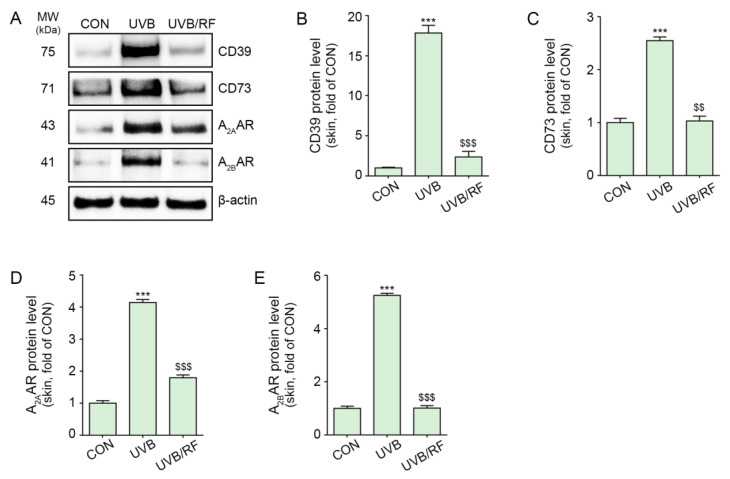
RF irradiation reduced CD39/CD73/A_2A_AR/A_2B_AR expression in UVB-exposed mouse skin. Mice were either not exposed or exposed to UVB nine times for 5 min each for 14 d (CON or UVB group), and then RF irradiation was applied (UVB/RF group). (**A**) The protein expression of CD39/CD73/A_2A_AR/A_2B_AR in UVB/RF-exposed mouse skin was validated using Western blot. (**B**–**E**) Quantitative graph for Western blot of (**A**). Data are presented as the mean ± SD of three independent experiments. ***, *p* < 0.001, second bar vs. first bar; $$, *p* < 0.01; $$$, *p* < 0.001, third bar vs. second bar (Mann–Whitney U test). A_2A_AR, A_2A_ adenosine receptor; A_2B_AR, A_2B_ adenosine receptor; CD39, ectonucleoside triphosphate diphosphohydrolase-1; CD73, 5′-nucleotidase; CM, conditioned media; CON, control; d, days; min, minutes; MW, molecular weight; RF, radiofrequency; UVB, ultraviolet B.

**Figure 3 ijms-24-05506-f003:**
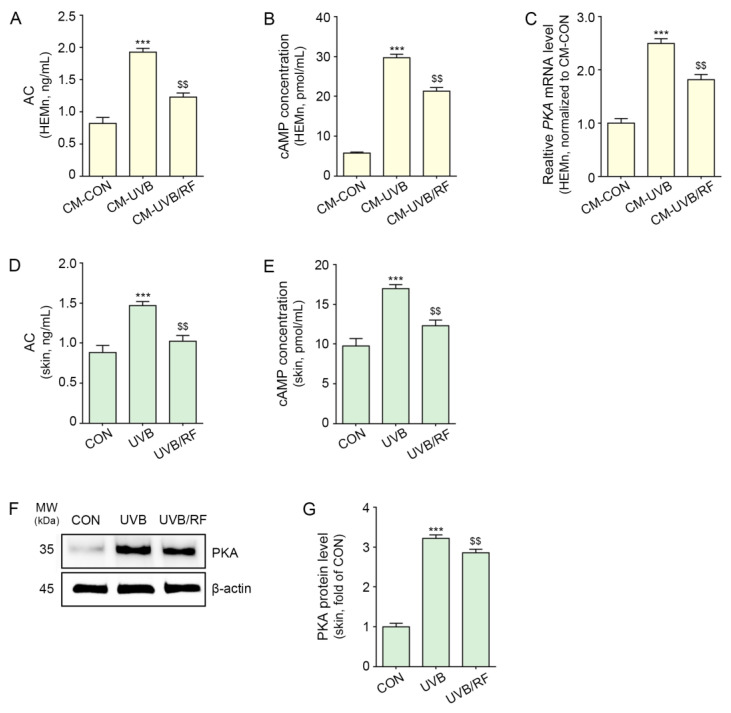
RF irradiation reduced the expression of AC/cAMP/PKA in CM-treated melanocytes and UVB-exposed mouse skin. (**A**–**C**) HEMn were treated with the supernatant of HEKn cells (CM-CON), UVB-exposed HEKn cells (CM-UVB), or UVB and RF-treated HEKn (CM-UVB/RF) for 48 h. (**A**,**B**) Changes in AC activity (**A**) and cAMP concentration (**B**) in CM-treated HEMn were measured by cAMP XP assay. (**C**) The mRNA levels of PKA in CM-treated HEMn were determined by qRT-PCR. (**D**–**G**) Mice were not exposed or exposed to UVB nine times for 5 min each for 14 d (CON or UVB group), and then RF was applied (UVB/RF group). (**D**,**E**) AC activity (**D**) and cAMP concentration (**E**) in UVB/RF-exposed mouse skin were measured using a cAMP XP assay. (**F**) The protein expression of PKA in UVB/RF-exposed mouse skin was measured using Western blot. (**G**) Quantitative graph of the Western blot of (**F**). Data are presented as the mean ± SD of three independent experiments. ***, *p* < 0.001, second bar vs. first bar; $$, *p* < 0.01, third bar vs. second bar (Mann–Whitney U test). AC, adenylate cyclase; cAMP, cyclic adenosine monophosphate; CM, conditioned media; CON, control; d, days; min, minutes; MW, molecular weight; PKA, cAMP-dependent protein kinase; h, hours; HEMn, human epidermal primary melanocytes; qRT-PCR, quantitative real-time polymerase chain reaction; RF, radiofrequency; UVB, ultraviolet B.

**Figure 4 ijms-24-05506-f004:**
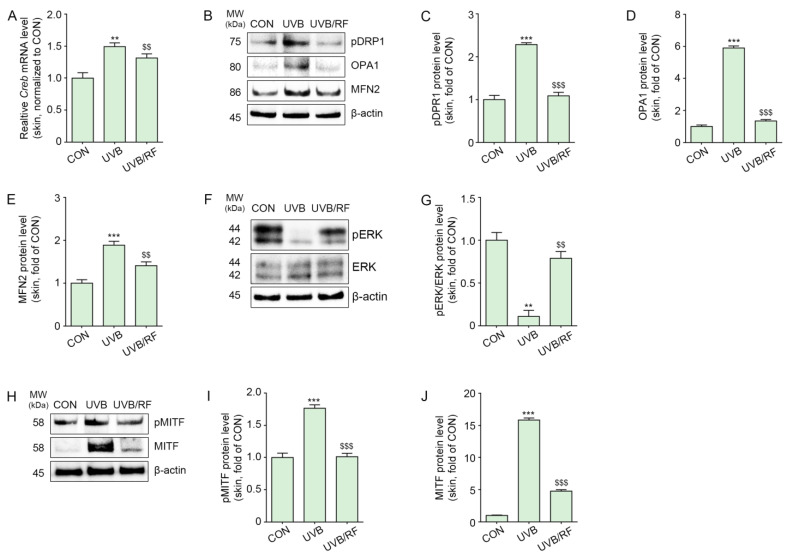
RF application decreased CREB/pDRP1/OPA1/MFN2/MITF in UVB-exposed mouse skin. Mice were either not exposed or exposed to UVB nine times for 5 min each for 14 d (CON or UVB group), and then RF was irradiated (UVB/RF group). (**A**) The mRNA levels of CREB in mouse skin were measured by qRT-PCR. (**B**) The protein expression of pDRP1/OPA/MFN2 in mouse skin was measured by Western blot. (**C**–**E**) Quantitative graph for Western blot of (**B**). (**F**) The protein expression of pERK1/2/ERK1/2 in mouse skin was determined by Western blot. (**G**) Quantitative graph of the Western blot of (**F**). (**H**) The protein expressions of pMITF and MITF in mouse skin were measured by Western blot. (**I**,**J**) Quantitative graph for Western blot of (**H**). Data are presented as the mean ± SD of three independent experiments. **, *p* < 0.01; ***, *p* < 0.001, second bar vs. first bar; $$, *p* < 0.01, $$$ < 0.001, third bar vs. second bar (Mann–Whitney U test). CON, control; CREB, cyclic adenosine monophosphate response element-binding protein; d, days; MFN2, mitofusin 2; min, minutes; MITF, microphthalmia-associated transcription factor; MW, molecular weight; OPA1, optic atrophy type 1; pDRP1, phosphorylation of dynamin-related protein 1; pMITF, phosphorylation of microphthalmia-associated transcription factor; qRT-PCR, quantitative real-time polymerase chain reaction; RF; RF, radiofrequency; UVB, ultraviolet B.

**Figure 5 ijms-24-05506-f005:**
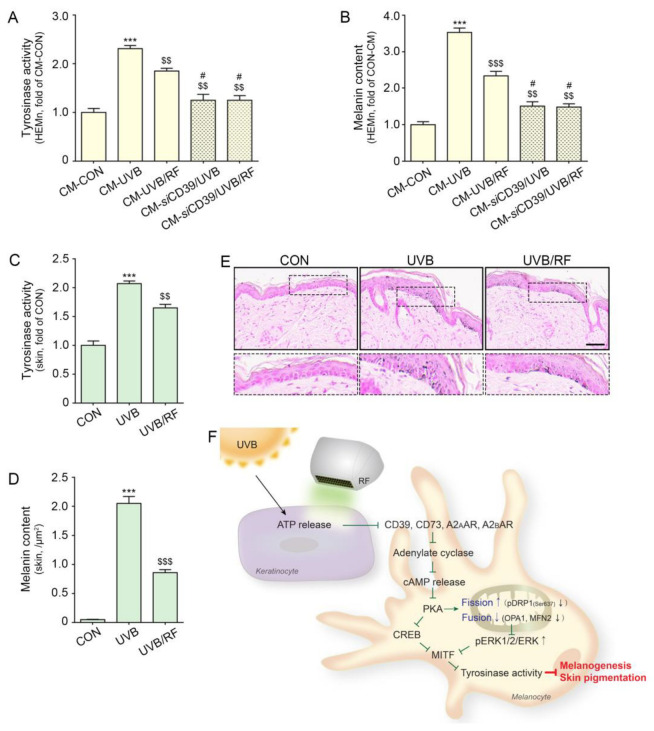
RF irradiation reduced tyrosinase activity and melanin accumulation through regulation of CD39. (**A**,**B**) HEMn were treated with the supernatant of HEKn cells (CM-CON), UVB-exposed HEKn (CM-CON), or UVB- and RF-treated HEKn (CM-UVB/RF) for 48 h. To confirm the regulation by CD39, after silencing CD39, HEMn were treated with CM-UVB (*si*CD39/CM-UVB) or CM-UVB/RF (CM-*si*CD39/UVB/RF). (**A**) The tyrosinase activity in CM-treated HEMn was measured using a tyrosinase activity assay. (**B**) Melanin accumulation was confirmed by a melanin assay in CM-treated HEMn. (**C**–**E**) Mice were either not exposed or exposed to UVB nine times for 5 min each for 14 d (CON or UVB group), and then RF was applied (UVB/RF group). (**C**) The tyrosinase activity in mouse skin was measured by tyrosinase activity assay. (**D**) Quantitative graph for Fontana Masson staining (**E**). (**E**) Melanin accumulation was determined by Fontana Masson staining in mouse skin (scale bar = 50 μm). (**F**) Summary. Data are presented as the mean ± SD of three independent experiments. ***, *p* < 0.001, second bar vs. first bar; $$, *p* < 0.01; $$$, *p* < 0.001, vs. second bar, #, *p* < 0.05, vs. third bar (Mann–Whitney U test). cAMP, cyclic adenosine monophosphate; CD39, ectonucleoside triphosphate diphosphohydrolase-1; CD73, 5′-nucleotidase; CM, conditioned media; CON, control; CREB, cyclic adenosine monophosphate response element-binding protein; MFN2, mitofusin 2; MITF, microphthalmia-associated transcription factor; MW, molecular weight; OPA1, optic atrophy type 1; pDRP1, phosphorylation of dynamin-related protein 1; RF, radiofrequency; siCD39, silencing CD39; UVB, ultraviolet B.

## Data Availability

All data is contained within the article.

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
