# Peer review of "Radiofrequency Irradiation Attenuated UVB-Induced Skin Pigmentation by Modulating ATP Release and CD39 Expression"

_ijms, 2023, doi:10.3390/ijms24065506_

Round 1
Reviewer 1 Report (Previous Reviewer 2)
The authors for the most parts adequately revised the manuscript. There are some pending corrections of stylistic and typographical deficiencies.
Also, in the introduction the authors could mention hormonal mechanisms regulating melanin pigmentation (Physiol Rev 84, 1155-1228, 2004), and diverse bioregulatory functions of melanin and melanogenesis in the skin (Frontiers in Oncology 2022;12. DOI: 10.3389/fonc.2022.842496).
Author Response
Response to Reviewer 1 Comments
The authors for the most parts adequately revised the manuscript.
Point 1. There are some pending corrections of stylistic and typographical deficiencies.
Answer 1. Thank you for your comment. We rechecked and revised the manuscript.
Point 2. Also, in the introduction the authors could mention hormonal mechanisms regulating melanin pigmentation (Physiol Rev 84, 1155-1228, 2004), and diverse bioregulatory functions of melanin and melanogenesis in the skin (Frontiers in Oncology 2022;12. DOI: 10.3389/fonc.2022.842496).
Answer 2. As your recommendation, we added references.
Introduction, Page 2, Line 43 in the revised manuscript The skin has protective roles as a barrier organ. The skin also has a neuroendocrine function that stimulates the central nervous, endocrine, and immune systems to coordinate body homeostasis [1–3]. External stimuli, such as ultraviolet (UV) radiation lead to the upregulation of various cytokines, uro-cortins, corticotropin-releasing hormone (CRH), proopiomelanocortin (POMC), and enkephalins, which further stimulate various skin responses, including melanogenesis [1–4]. |
Introduction, Page 15, Line 567 in the revised manuscript 2. Slominski, A.; Tobin, D.J.; Shibahara, S.; Wortsman, J. Melanin pigmentation in mammalian skin and its hormonal regulation. Physiol Rev 2004, 84, 1155–1228. 3. Slominski, R.M.; Sarna, T.; PÅ‚onka, P.M.; Raman, C.; Brożyna, A.A.; Slominski, A.T. Melanoma, Melanin, and Melanogenesis: The Yin and Yang Relationship. Front Oncol 2022, 12, 842496. |

Reviewer 2 Report (Previous Reviewer 1)
All my comments and requests were addressed by the authors. Thank you.
Author Response
Thank you for your review.
This manuscript is a resubmission of an earlier submission. The following is a list of the peer review reports and author responses from that submission.
Round 1
Reviewer 1 Report
The authors investigated radiofrequency as a potential therapeutic method against unwanted pigmentation. It is a continuation of a previously published paper. I only have a few small comments:
-Either Figures 1 and 2 are missing or the numbering of the presented figures must be changed.
-line 164-165: the authors state that the decreasing effect of RF disappeared when CD39 was silenced. I think this wording is incorrect. According to figure S1, the silencing of CD39 was done before RF-irradiation. Since siCD39 already decreased pigmentation events, it is no surprise that the decreasing effect of RF cannot be seen. There is nothing to decrease, is there?
-in the figure captions, the authors mention using ELISA for tyrosinase activity. In the methods the list an enzyme inhibition assay for tyrosinase. Unless I am wrong, this is enzyme inhibition assay is not an ELISA. the authors should change/remove ELISA from the captions because it implies that the authors looked at tyrosinase expression, not activity.
-From my point of view, the first part of the Discussion until line 215 belongs to the Introduction.
-In general in the Discussion I miss references to the figures. If the authors could add those, it would simplify reading of the discussion.
-line 259-260: again see my comment above for the decreasing effect. I think this should be rephrased.
-again same comment for line 305-306 about the decreasing effect.
-4.2.1.: What device did you use for UVB irradiation? What wavelength exactly and which fluency/dosage?
-4.2.4. line 340: ...an 1 N. I think there is something missing here?
-4.3. line 351-353: Was this the same UVB dosage as for the in vitro tests? Again, what device was used?
-4.8.1. line 446: how were the slides coated?
Author Response
Response to Reviewer 1 Comments
The authors investigated radiofrequency as a potential therapeutic method against unwanted pigmentation. It is a continuation of a previously published paper. I only have a few small comments:
Point 1. Either Figures 1 and 2 are missing or the numbering of the presented figures must be changed.
Answer 1. We are sorry that the omission of Figures 1 and 2 caused confusion. We have added Figures 1 and 2.
Result, Page 4, Line 127 in revised manuscript 2.1. RF irradiation decreased ATP release from UV-radiated keratinocytes First, we evaluated whether RF irradiation decreased ATP release from UVB-exposed keratinocytes. After human keratinocytes were subjected to UVB radiation, RF irradiation was performed. Forty-eight hours after RF irradiation, ATP levels in the supernatant of keratinocyte were measured (Figure S1A). The ATP level was significantly increased by UVB radiation and significantly de-creased by RF treatment (Figure 1A). |
Result, Page 4, Line 148 in revised manuscript 2.2. RF decreased the expression of CD39, CD73 and the A2A/A2BARs in melanocytes and UV-irradiated animal skin Next, we evaluated whether RF irradiation decreased the expressions of CD39, CD73, and A2A/A2BARs in melanocytes by decreasing ATP release from keratinocytes. The conditioned media (CM) obtained from CON-, UVB-, or UVB and RF-irradiated keratinocytes were used to treat human melanocytes (Figure S1B). The expression of CD39 and CD73 in melanocytes was significantly increased with treatment of CM from UVB-irradiated keratinocytes (CM-UVB) and significantly decreased with treatment of CM from UVB- and RF-irradiated keratinocytes (CM-UVB/RF) (Figures 1B–1D). The expression of the A2A/A2BARs in melanocytes significantly increased under treatment with CM-UVB and significantly decreased under treatment with CM-UVB/RF (Figures 1B, 1E, and 1F). The expression of CD39 and CD73 was also increased by UVB radiation in the skin of animals subjected to UVB radiation compared with that of control animals that did not receive UVB radiation. The expressions of CD39 and CD73 were significantly decreased by RF irradiation (Figures 2A–2C). Moreover, the expression of A2A/A2BARs was increased in the skin of animals subjected to UVB radiation and decreased by RF irradiation (Figure 2A, 2D, and 2E). |
Figure 1, Page 4, Line 134 in revised manuscript
Figure 1. CD39/CD73/A2AAR/A2BAR reduction in melanocytes due to ATP release from UVB/RF-exposed keratinocytes. HEKn were not exposed or exposed to UVB for 5 min (CON or UVB group), and then RF was irradiated after exposure to UVB (UVB/RF group). (A) ATP release in the CON group, UVB group and UVB/RF group was verified using an ATP release assay. (B–F) HEMn were treated with a supernatant of HEKn (CM-CON), UVB-exposed HEKn (CM-UVB), or UVB and RF-treated HEKn (CM-UVB/RF) for 48 h. (B) The protein expression of CD39/CD73/A2AAR/A2BAR in CM-treated HEMn was measured by western blot. (C–F) Quantitative graph for western blot of (B). Data are presented as the mean ± SD of three independent experiments. **, p < 0.01; ***, p < 0.001, second bar vs. first bar; $$, p < 0.01, third bar vs. second bar (Mann–Whitney U test). A2AAR, A2A adenosine receptor; A2BAR, A2B adenosine receptor; ATP, adenosine triphosphate; CD39, ectonucleoside triphosphate diphosphohydrolase-1; CD73, 5′-nucleotidase; CM, conditioned media; CON, control; HEKn, human epidermal primary keratinocytes; h, hours; HEMn, human epidermal primary melanocytes; MW, molecular weight; RF, radiofrequency; UVB, ultraviolet B. |
Figure 2, Page 5, Line 169 in revised manuscript
Figure 2. RF irradiation reduced CD39/CD73/A2AAR/A2BAR expression in UVB-exposed mouse skin. Mice were either not exposed or exposed to UVB 9 times for 5 min each for 14 d (CON or UVB group), and then RF irradiation was applied (UVB/RF group). (A) The protein expression of CD39/CD73/A2AAR/A2BAR in UVB/RF-exposed mouse skin was validated using western blot. (B–E) Quantitative graph for western blot of (A). Data are presented as the mean ± SD of three independent experiments. ***, p < 0.001, second bar vs. first bar; $$, p < 0.01; $$$, p < 0.001, third bar vs. second bar (Mann–Whitney U test). A2AAR, A2A adenosine receptor; A2BAR, A2B adenosine receptor; CD39, ectonucleoside triphosphate diphosphohydrolase-1; CD73, 5′-nucleotidase; CM, conditioned media; CON, control; d, days; min, minutes; MW, molecular weight; RF, radiofrequency; UVB, ultraviolet B. |
Point 2. line 216-217: the authors state that the decreasing effect of RF disappeared when CD39 was silenced. I think this wording is incorrect. According to figure S1, the silencing of CD39 was done before RF-irradiation. Since siCD39 already decreased pigmentation events, it is no surprise that the decreasing effect of RF cannot be seen. There is nothing to decrease, is there?
Answer 2. As your recommendation, we changed the sentence like below;
Result, Page 8, Line 245 in revised manuscript Next, we evaluated whether RF irradiation decreased melanin synthesis. Tyrosinase activity in melanocytes was increased by the administration of CM-UVB. Upon silencing CD39, tyrosinase activity was not increased by the administration of CM-UVB compared with the CON. The increased tyrosinase activity caused by the administration of CM-UVB was significantly decreased by CM-UVB/RF treatment. However, among silencing CD39 melanocyte, tyrosinase activity was not different when either CM-UVB or CM-UVB/RF were treated (Figure 5A). The melanin content in melanocytes was increased by the administration of CM-UVB. Upon silencing CD39, melanin content was not increased by the administration of CM-UVB compared with that of the CON. The increased melanin content by administration of CM-UVB was significantly decreased by CM-UVB/RF administration. However, among silencing CD39 melanocyte, melanin content was not different following treatment by either CM-UVB or CM-UVB/RF (Figure 5B). |
Point 3. in the figure captions, the authors mention using ELISA for tyrosinase activity. In the methods the list an enzyme inhibition assay for tyrosinase. Unless I am wrong, this is enzyme inhibition assay is not an ELISA. the authors should change/remove ELISA from the captions because it implies that the authors looked at tyrosinase expression, not activity.
Answer 3. As your recommendation, we changed the captions like below;
Figure 5, Page 9, Line 266 in revised manuscript Figure 5. RF irradiation reduced tyrosinase activity and melanin accumulation through regulation of CD39. (A and B) HEMn were treated with the supernatant of HEKn cells (CM-CON), UVB-exposed HEKn (CM-CON), or UVB- and RF-treated HEKn (CM-UVB/RF) for 48 h. To confirm the regulation by CD39, after silencing CD39, HEMn were treated with CM-UVB (siCD39/CM-UVB) or CM-UVB/RF (CM-siCD39/UVB/RF). (A) The tyrosinase activity in CM-treated HEMn was measured using a tyrosinase activity assay. (B) Melanin accumulation was confirmed by a melanin assay in CM-treated HEMn. (C–E) Mice were either not exposed or exposed to UVB 9 times for 5 min each for 14 d (CON or UVB group), and then RF was applied (UVB/RF group). (C) The tyrosinase activity in mouse skin was measured by tyrosinase activity assay. (D) Quantitative graph for Fontana Masson staining (E). (E) Melanin accumulation was determined by Fontana Masson staining in mouse skin. (F) Summary. Data are presented as the mean ± SD of three independent experiments. ***, p < 0.001, second bar vs. first bar; $$, p < 0.01; $$$, p < 0.001, vs. second bar, #, p < 0.05, vs. third bar (Mann–Whitney U test). cAMP, cyclic adenosine monophosphate; CD39, ectonucleoside triphosphate diphosphohydrolase-1; CD73, 5′-nucleotidase; CM, conditioned media; CON, control; CREB, cyclic adenosine monophosphate response element-binding protein; MFN2, mitofusin 2; MITF, microphthalmia‑associated transcription factor; MW, molecular weight; OPA1, optic atrophy type 1; pDRP1, phosphorylation of dynamin-related protein 1; RF, radiofrequency; siCD39, silencing CD39; UVB, ultraviolet B. |
Point 4. From my point of view, the first part of the Discussion until line 259 belongs to the Introduction.
Answer 4. As your recommendation, we moved those sentences to the Introduction and revised them.
Point 5. In general in the Discussion I miss references to the figures. If the authors could add those, it would simplify reading of the discussion.
Answer 5. As your recommendation, we added references to the figures in the Discussion.
Discussion, Page 10, Line 299 in revised manuscript The purpose of our study was to evaluate whether RF could decrease UV-B induced melanogenesis via decreasing ATP release from keratinocytes. Thus, we designed an in vitro model in which CM from UVB-irradiated keratinocytes was administered to melanocytes. First, we sought to evaluate whether UV radiation increased ATP release from keratinocytes. ATP levels in the supernatant from UVB-irradiated keratinocytes increased compared with those of control keratinocytes (Figure 1A). Next, we evaluated how elevated ATP secretion from keratinocytes affects melanocytes to increase melanogenesis. Upon treating melanocytes with CM from UVB-irradiated keratinocytes, the expressions of CD39, CD73, and A2A/A2BARs increased (Figures 1B–1F). Moreover, AC activity and cAMP levels were increased by CM from UVB-irradiated keratinocytes. It seemed that UVB radiation increased ATP release from keratinocytes, and released extracellular ATP was converted into adenosine by CD39 and CD73, which stimulated A2A/A2BARs. The upregulation of A2A/A2BARs led to increased AC activity and increased cAMP levels in melanocytes (Figures 3A and 3B). Moreover, increased cAMP was accompanied by an increased expression of PKA in melanocytes treated with CM from UVB-irradiated keratinocytes (Figure 3C). |
Discussion, Page 10, Line 316 in revised manuscript Our results also show that increased tyrosinase activity and melanin contents in melanocytes via treatment with CM from UVB radiated keratinocytes. When CD39 was silenced by siCD39, tyrosinase activity and melanin content were not increased by treatment with CM from UVB radiated keratinocytes (Figures 5A–5B). This result suggests that melanogenesis triggered in melanocytes by ATP secretion from keratinocytes could be mediated by CD39. To the best of our knowledge, this is the first study that shows the role of CD39 and ARs in UV radiation-induced melanogenesis. UV radiation also led to increased expressions of CD39, CD73, and A2A/A2BARs, as well as AC activity, cAMP levels, and PKA expression in animal skin (Figures 2 and 3D–G). Moreover, melanogenesis-related factors such as CREB and MITF were increased in UV-irradiated animal skin (Figures 4A, 4H, and 4I). |
Discussion, Page 11, Line 355 in revised manuscript Since the phosphorylation of DRP1 at Ser637 via cAMP-mediated PKA activation is known to downregulate DRP1 and eventually lead to mitochondrial elongation [27], we evaluated whether UV-induced DRP1 phosphorylation at Ser637 modulates mitochondrial dynamics via cAMP-mediated PKA activation. UV-irradiated animal skin had an increased expression of phosphorylated DRP at Ser637, and RF irradiation decreased this expression (Figures 4B and 4C). We did not directly evaluate mitochondrial morphology using any imaging technique. However, mitochondrial fusion-related proteins such as OPA1 and MFN2 were increased by UV radiation in animal skin and decreased by RF irradiation (Figures 4B, 4D, and 4E). Although we did not observe mitochondrial elongation with imaging, we could infer that increased phosphorylation of DRP at Ser637 affects mitochondrial dynamics. Since mitochondrial fission activates ERK and degrades MITF [31], we also evaluated ERK changes after RF irradiation. UVB radiation decreased the pERK1/2/ERK1/2 ratio, which was increased by RF irradiation (Figure 4F–4G). |
Point 6.ine 295-296: again see my comment above for the decreasing effect. I think this should be rephrased.
Answer 6. We are sorry that the omission of Figures 1 and 2 caused confusion. In Figure 1, we showed that ATP release was increased by UVB radiation in the keratinocytes and RF leads to decreased ATP release. In Figures 1 and 2, we showed that UVB leads to increased CD39, CD73, A2A/A2B adenosine receptors in either melanocytes or animal skin. Thus, the sentence that you mentioned, ‘RF irradiation decreased ATP levels in keratinocytes subjected to UVB radiation’, appears to be valid for explaining our results.
Point 7. again same comment for line 339-340 about the decreasing effect.
Answer 7. We are sorry that the omission of Figures 1 and 2 caused confusion. In Figure 1, we showed that ATP release was increased by UVB radiation in the keratinocytes and RF leads to decreased ATP release. In Figures 1 and 2, we showed that UVB leads to increased CD39, CD73, A2A/A2B adenosine receptors in either melanocytes or animal skin. Thus, the sentence that you mentioned, ‘RF irradiation decreased ATP levels in keratinocytes subjected to UVB radiation’, appears to be valid for explaining our results.
Point 8. 4.2.1.: What device did you use for UVB irradiation? What wavelength exactly and which fluency/dosage?
Answer 8. As your recommendation, we added UVB information.
Material and Method, Page 11, Line 361 in revised manuscript HEKn (ATCC, Manassas, VA, USA) were cultivated in a Dermal Cell Basal Medium (ATCC) containing a keratinocyte growth kit (ATCC) in an incubator at 37 °C with 5% CO2. When HEKn reached 70% confluence, RF (2 MHz, 10 W/100 ms) was applied after exposure to UVB (200 mJ/cm2) for 5 min. UVB used an instrument with a peak wavelength of 306 nm (G15T8E; SANKYO DENKI, Yokohama, Japan). RF-treated HEKn were cultured in an incubator at 37 °C with 5% CO2 for 48 h, and the cell lysate and supernatant (conditioned media, CM) were obtained (Figure S1A). |
Point 9. 4.2.4. line 340: ...an 1 N. I think there is something missing here?
Answer 9. As your recommendation, we revised that sentence.
Point 10. 4.3. line 351-353: Was this the same UVB dosage as for the in vitro tests? Again, what device was used?
Answer 10. We used the same device for in vitro and in vivo.
Point 11. 4.8.1. line 446: how were the slides coated?
Answer 11. We used silane-coated slides. The coated slides were purchased from MUTO PURE CHEMICALS CO., LTD. The information is displayed in the manuscript.

Reviewer 2 Report
The topic is of interest, however, the paper requires significant revisions and clarifications. While methodology is sound, with some exception, the introduction and discussion lacks the clarity, perhaps due to problems in English presentation. Also there are factual errors as relates to mechanism regulating melanogenesis. Also clarification on RF are needed for readers to understand its role.
Please in the introduction described what do you mean by radiofrequency. Describe properties and concept. The same in the MM with description of physical effects and wavelength. This is important for unprepared reader to understand the presented concept.
Factual deficiencies in the introduction include limtited description how melanogenesis is regulated. There are many pathways and mechanisms regulating melanogenesis in addition cAMP signaling (Physiol Rev 84, 1155-1228, 2004; Pigment Cell Melanoma Res 25, 14-27, 2012). This requires corrections and better overview.
What do you mean by melanosomes migrate to keratinocytes. There is no migration by active transport or secretion through precise mechanisms. Please improve your knowledge on this topic.
As relates to UVR, an important subject of this investigation, the readers would appreciate information on diverse homeostatic actions of the UVR on cutaneous and systemic homeostasis (Endocrinology 159(5), 1992-2007, 2018).
In results, measuring MITF only at RNA level may not be informative, since it is a master regulator of melanogenesis of which activity is regualted predominantly at posttranslational level, phosphorylation.
Again provide pertinent information on RF
Provide microscopic bars for micrographs
Also is ATP release due to its increase production and regulated relelease or due to diffusion through damaged membranes
In the induction, again you provide simplistic mechanism of regulation of melanogenesis through POMC-derived MSH acting on MC1. This is not fully correct, since such effects are context dependent, as documented by lak of effect of knocking out POMC in C57BL6 mice on eumelanin production as reported in 2005 in Endocrinology.
Since authors discuss melanogenesis the readers would appreciate mentioning pleiotropic effects of this process and pigment on cellular level (Frontiers in Oncology 2022;12. DOI: 10.3389/fonc.2022.842496).
Since authors propose complex interactions on epidermal level (fig. 5F), they are encourage to mention diverse neuropeptides activities in the epidermis (American Journal of Physiology-Cell Physiology 2022 323:6, C1757-C1776)
Author Response
Response to Reviewer 2 Comments
The topic is of interest, however, the paper requires significant revisions and clarifications. While methodology is sound, with some exception, the introduction and discussion lacks the clarity, perhaps due to problems in English presentation. Also there are factual errors as relates to mechanism regulating melanogenesis. Also clarification on RF are needed for readers to understand its role.
Point 1. Please in the introduction described what do you mean by radiofrequency. Describe properties and concept. The same in the MM with description of physical effects and wavelength. This is important for unprepared reader to understand the presented concept.
Answer 1. As your recommendation, we added explanation of RF in the Introduction part as below;
Introduction, Page 3, Line 104 in revised manuscript Radiofrequency (RF) is a form of electromagnetic energy that generates molecular agitation in tissues [35]. Moreover, a conducted electric current caused by RF is transformed into heat [35]. Since heat generated by RF stimulates collagen synthesis by increasing the expression of various heat shock proteins (HSPs), such as HSP47 and HSP70, RF has been used for skin rejuvenation [36,37]. |
Moreover, we added information on RF wavelength in the MM as below;
Materials and Method, Page 10, Line 348 in revised manuscript 4.1. Radiofrequency irradiation system A radiofrequency device (POTENZA, Jeisys, Seoul, Korea) was used in this study. It has an impedance checking and feedback system which was used to determine the compensation value by automatically measuring impedance. The most common configurations of electrode systems are monopolar, bipolar, and multipolar, including fractional systems where this effect is achieved by the superposition of RF current paths between paired electrodes. Bipolar radiofrequency energy was delivered to the skin surface and epidermis via a noninvasive electrode tip, SFA tip. The tip consisted of 192 electrodes with a diameter of 0.2 mm arranged at a spacing of 1 mm, and it is designed to deliver RF energy at a shallow depth. The RF conditions was treated 10W, 100 ms, and 2 pulse. Additionally, the frequency was 2 MHz, and the wavelength is inversely proportional to the frequency [53]. |
Point 2. Factual deficiencies in the introduction include limtited description how melanogenesis is regulated. There are many pathways and mechanisms regulating melanogenesis in addition cAMP signaling (Physiol Rev 84, 1155-1228, 2004; Pigment Cell Melanoma Res 25, 14-27, 2012). This requires corrections and better overview.
Answer 2. As your recommendation, we rewrote the introduction like below;
Introduction, Page 2, Line 50 in revised manuscript Melanogenesis is a complex process that produces melanin in the melanocytes. The initial steps of melanogenesis involve the hydroxylation of phenylalanine to L-tyrosine [3–5]. L-tyrosine is then hydroxylated to L-dihydroxyphenylalanine (L-DOPA) by tyrosinase. Tyrosinase further oxidizes L-DOPA to L-DOPAquinone [3–5]. Depending on the presence of cysteine, L-DOPAquinone changes into either yellow-to-reddish pheomelanin or brown-to-blackish eumelanin [6–8]. Without cysteine, L-DOPAquinone is changed into DOPAchrome by the tyrosinase‑related protein (TRP)‑1 and TRP‑2, which is then further synthesized to eumelanin [6–8]. UV upregulates tumor suppressor protein p53, which further stimulates POMC. POMC are cleaved into adrenocorticotropic hormone (ACTH), α-melanocyte-stimulating hormone (MSH), β-MSH, and γ-MSH [9–14]. Secreted α-MSH binds to a melanocortin receptor (MC1R) in melanocytes, which promotes the dissociation of the α subunit and eventually upregulates the activity of adenylate cyclase (AC) [15]. ACs generate the second messenger cyclic adenosine monophosphate (cAMP) from adenosine triphosphate (ATP) [15]. cAMP leads to the increased activity of the cAMP-dependent protein kinase (PKA) and the transcription factor cAMP response element-binding protein (CREB), which sequentially increases the expression of microphthalmia-associated transcription factor (MITF) [16]. MITF is the main regulator of melanogenesis, modulating the survival, proliferation, and growth of melanocytes [17]. MITF also stimulates tyrosinase, TRP‑1, and TRP‑2 and eventually increases melanogenesis [17]. Aside from the cAMP-dependent pathway, Wnt/β-catenin, ERK/MAPK, and nitric oxide/cyclic guanosine monophosphate (cGMP) pathway lead to upregulation of MITF in melanogenesis [18]. |
Point 3. What do you mean by melanosomes migrate to keratinocytes. There is no migration by active transport or secretion through precise mechanisms. Please improve your knowledge on this topic.
Answer 3. The purpose of present study was to evaluate whether RF downregulated melanogenesis via decreasing the cAMP-dependent pathway, not via decreasing melanosome transfer to keratinocytes. Thus, we deleted that sentence that you addressed in order to improve readability.
Point 4. As relates to UVR, an important subject of this investigation, the readers would appreciate information on diverse homeostatic actions of the UVR on cutaneous and systemic homeostasis (Endocrinology 159(5), 1992-2007, 2018).
Answer 4. We added information related to the diverse homeostatic actions of the UVR on cutaneous in the Introduction part as below;
Introduction, Page 2, Line 43 in revised manuscript The skin has protective roles as a barrier organ. The skin also has a neuro-endocrine function that stimulates the central nervous, endocrine, and immune systems to coordinate body homeostasis [1]. External stimuli, such as ultraviolet (UV) radiation lead to the upregulation of various cytokines, urocortins, corticotropin-releasing hormone (CRH), proopiomelanocortin (POMC), and enkephalins, which further stimulate various skin responses, including melanogenesis [1,2]. |
Point 5. In results, measuring MITF only at RNA level may not be informative, since it is a master regulator of melanogenesis of which activity is regualted predominantly at posttranslational level, phosphorylation.
Answer 5. As your recommendation, we added MITF and pMITF protein levels using western blot.
Result, Page 7, Line 225 in revised manuscript Additionally, pMITF and MITF expression levels increased in the skin of animals subjected to UVB radiation and decreased by RF irradiation (Figures 4H–4J). |
Figure 4, Page 7, Line 228 in revised manuscript
Figure 4. RF application decreased CREB/pDRP1/OPA1/MFN2/MITF in UVB-exposed mouse skin. Mice were either not exposed or exposed to UVB 9 times for 5 min each for 14 d (CON or UVB group), and then RF was irradiated (UVB/RF group). (A) The mRNA levels of CREB in mouse skin were measured by qRT‒PCR. (B) The protein expression of pDRP1/OPA/MFN2 in mouse skin was measured by western blot. (C–E) Quantitative graph for western blot of (B). (F) The protein expression of pERK1/2/ERK1/2 in mouse skin was determined by western blot. (G) Quantitative graph of the western blot of (F). (H) The protein expression of pMITF and MITF in mouse skin were measured by western blot. (I–J) Quantitative graph for western blot of (H). Data are presented as the mean ± SD of three independent experiments. **, p < 0.01; ***, p < 0.001, second bar vs. first bar; $$, p < 0.01, third bar vs. second bar (Mann–Whitney U test). CON, control; CREB, cyclic adenosine monophosphate response element-binding protein; d, days; MFN2, mitofusin 2; min, minutes; MITF, microphthalmia‑associated transcription factor; MW, molecular weight; OPA1, optic atrophy type 1; pDRP1, phosphorylation of dynamin-related protein 1; pMITF, phosphorylation of microphthalmia‑associated transcription factor; qRT-PCR, quantitative real-time polymerase chain reaction; RF; RF, radiofrequency; UVB, ultraviolet B. |
Point 6. Again provide pertinent information on RF
Answer 6. As your recommendation, we added explanation of RF in the introduction part as below;
Introduction, Page 3, Line 104 in revised manuscript Radiofrequency (RF) is a form of electromagnetic energy that generates molecular agitation in tissues [35]. Moreover, a conducted electric current caused by RF is transformed into heat [35]. Since heat generated by RF stimulates collagen synthesis by increasing the expression of various heat shock proteins (HSPs), such as HSP47 and HSP70, RF has been used for skin rejuvenation [36,37]. |
Moreover, we added information on RF wavelength in the MM as below;
Materials and Method, Page 10, Line 348 in revised manuscript 4.1. Radiofrequency irradiation system A radiofrequency device (POTENZA, Jeisys, Seoul, Korea) was used in this study. It has an impedance checking and feedback system which was used to determine the compensation value by automatically measuring impedance. The most common configurations of electrode systems are monopolar, bipolar, and multipolar, including fractional systems where this effect is achieved by the superposition of RF current paths between paired electrodes. Bipolar radiofrequency energy was delivered to the skin surface and epidermis via a noninvasive electrode tip, SFA tip. The tip consisted of 192 electrodes with a diameter of 0.2 mm arranged at a spacing of 1 mm, and it is designed to deliver RF energy at a shallow depth. The RF conditions was treated 10W, 100 ms, and 2 pulse. Additionally, the frequency was 2 MHz, and the wavelength is inversely proportional to the frequency [53]. |
Point 7. Provide microscopic bars for micrographs.
Answer 7. As your recommendation, we added microscopic bars.
Figure 5, Page 10, Line 265 in revised manuscript
Figure 5. RF irradiation reduced tyrosinase activity and melanin accumulation through regulation of CD39. (A and B) HEMn were treated with the supernatant of HEKn cells (CM-CON), UVB-exposed HEKn (CM-CON), or UVB- and RF-treated HEKn (CM-UVB/RF) for 48 h. To confirm the regulation by CD39, after silencing CD39, HEMn were treated with CM-UVB (siCD39/CM-UVB) or CM-UVB/RF (CM-siCD39/UVB/RF). (A) The tyrosinase activity in CM-treated HEMn was measured using a tyrosinase activity assay. (B) Melanin accumulation was confirmed by a melanin assay in CM-treated HEMn. (C–E) Mice were either not exposed or exposed to UVB 9 times for 5 min each for 14 d (CON or UVB group), and then RF was applied (UVB/RF group). (C) The tyrosinase activity in mouse skin was measured by tyrosinase activity assay. (D) Quantitative graph for Fontana Masson staining (E). (E) Melanin accumulation was determined by Fontana Masson staining in mouse skin (scale bar = 50 μm). (F) Summary. Data are presented as the mean ± SD of three independent experiments. ***, p < 0.001, second bar vs. first bar; $$, p < 0.01; $$$, p < 0.001, vs. second bar, #, p < 0.05, vs. third bar (Mann–Whitney U test). cAMP, cyclic adenosine monophosphate; CD39, ectonucleoside triphosphate diphosphohydrolase-1; CD73, 5′-nucleotidase; CM, conditioned media; CON, control; CREB, cyclic adenosine monophosphate response element-binding protein; MFN2, mitofusin 2; MITF, microphthalmia‑associated transcription factor; MW, molecular weight; OPA1, optic atrophy type 1; pDRP1, phosphorylation of dynamin-related protein 1; RF, radiofrequency; siCD39, silencing CD39; UVB, ultraviolet B. |
Point 8. Also is ATP release due to its increase production and regulated relelease or due to diffusion through damaged membranes
Answer 8. In this study, we evaluated the ATP level in the supernatants from keratinocytes that radiated to UVB or UVB and RF. Since we did not evaluate the releasing mechanism of ATP from keratinocytes, we can only say that ATP release was increased by UV-B radiation in our study. Maybe all of the recommended pathways that increased production, also increased release due to damaged membranes, potentially increasing extracellular ATP.
Point 9. In the induction, again you provide simplistic mechanism of regulation of melanogenesis through POMC-derived MSH acting on MC1. This is not fully correct, since such effects are context dependent, as documented by lak of effect of knocking out POMC in C57BL6 mice on eumelanin production as reported in 2005 in Endocrinology.
Answer 9. As your recommendation, we added the melanogenesis mechanism in the Introduction part as below;
Introduction, Page 2, Line 50 in revised manuscript Melanogenesis is a complex process that produces melanin in the melanocytes. The initial steps of melanogenesis involve the hydroxylation of phenylalanine to L-tyrosine [3–5]. L-tyrosine is then hydroxylated to L-dihydroxyphenylalanine (L-DOPA) by tyrosinase. Tyrosinase further oxidizes L-DOPA to L-DOPAquinone [3–5]. Depending on the presence of cysteine, L-DOPAquinone changes into either yellow-to-reddish pheomelanin or brown-to-blackish eumelanin [6–8]. Without cysteine, L-DOPAquinone is changed into DOPAchrome by the tyrosinase‑related protein (TRP)‑1 and TRP‑2, which is then further synthesized to eumelanin [6–8]. UV upregulates tumor suppressor protein p53, which further stimulates POMC. POMC are cleaved into adrenocorticotropic hormone (ACTH), α-melanocyte-stimulating hormone (MSH), β-MSH, and γ-MSH [9–14]. Secreted α-MSH binds to a melanocortin receptor (MC1R) in melanocytes, which promotes the dissociation of the α subunit and eventually upregulates the activity of adenylate cyclase (AC) [15]. ACs generate the second messenger cyclic adenosine monophosphate (cAMP) from adenosine triphosphate (ATP) [15]. cAMP leads to the increased activity of the cAMP-dependent protein kinase (PKA) and the transcription factor cAMP response element-binding protein (CREB), which sequentially increases the expression of microphthalmia-associated transcription factor (MITF) [16]. MITF is the main regulator of melanogenesis, modulating the survival, proliferation, and growth of melanocytes [17]. MITF also stimulates tyrosinase, TRP‑1, and TRP‑2 and eventually increases melanogenesis [17]. Aside from the cAMP-dependent pathway, Wnt/β-catenin, ERK/MAPK, and nitric oxide/cyclic guanosine monophosphate (cGMP) pathway lead to upregulation of MITF in melanogenesis [18]. |
Point 10. Since authors discuss melanogenesis the readers would appreciate mentioning pleiotropic effects of this process and pigment on cellular level (Frontiers in Oncology 2022;12. DOI: 10.3389/fonc.2022.842496).
Answer 10. As your recommendation, we mentioned the pleiotropic effect of melanogenesis in the Discussion like below;
Discussion, Page 11, Line 286 in revised manuscript Melanin in the skin, as a photoprotective pigment, absorbs UV and acts as scavenger against reactive oxygen/nitrogen species (ROS/RNS) [40]. Proper melanogenesis after UV exposure is essential for skin protection; however, the overproduction of melanin causes various cosmetic problems, such as senile lentigines, post-inflammatory hyperpigmentation, freckles, and dots [41]. |
Point 11. Since authors propose complex interactions on epidermal level (fig. 5F), they are encourage to mention diverse neuropeptides activities in the epidermis (American Journal of Physiology-Cell Physiology 2022 323:6, C1757-C1776)
Answer 11. As your recommendation, we added neuropeptides activity in the epidermis like below;
Introduction, Page 2, Line 43 in revised manuscript The skin has protective roles as a barrier organ. The skin also has a neuro-endocrine function that stimulates the central nervous, endocrine, and immune systems to coordinate body homeostasis [1]. External stimuli, such as ultraviolet (UV) radiation lead to the upregulation of various cytokines, urocortins, corticotropin-releasing hormone (CRH), proopiomelanocortin (POMC), and enkephalins, which further stimulate various skin responses, including melanogenesis [1,2]. |
